# Preparation, Modification, and Application of Biochar in the Printing Field: A Review

**DOI:** 10.3390/ma16145081

**Published:** 2023-07-19

**Authors:** Xin Li, Jinyu Zeng, Shuai Zuo, Saiting Lin, Guangxue Chen

**Affiliations:** State Key Laboratory of Pulp and Paper Engineering, South China University of Technology, Guangzhou 510640, China

**Keywords:** biochar, application, high value, printing field

## Abstract

Biochar is a solid material enriched with carbon produced by the thermal transformation of organic raw materials under anoxic or anaerobic conditions. It not only has various environmental benefits including reducing greenhouse gas emissions, improving soil fertility, and sequestering atmospheric carbon, but also has the advantages of abundant precursors, low cost, and wide potential applications, thus gaining widespread attention. In recent years, researchers have been exploring new biomass precursors, improving and developing new preparation methods, and searching for more high-value and meaningful applications. Biochar has been extensively researched and utilized in many fields, and recently, it has also shown good industrial application prospects and potential application value in the printing field. In such a context, this article summarizes the typical preparation and modification methods of biochar, and also reviews its application in the printing field, to provide a reference for future work.

## 1. Introduction

Carbon is an element that exists widely on the earth and has been used for a long time, playing an essential role in the progress of human civilization [1]. With the progress of society and the development of science and technology, through the continuous in-depth understanding and study of carbon elements, researchers have invented many new carbon materials (fullerenes, graphite, graphene, carbon nanotubes [2], carbon particles, carbon scaffolds [3], etc.) and have explored and studied them extensively. However, they are mostly synthesized using non-renewable petroleum and coal carbon products, and the synthesis process usually also requires harsh and energy-intensive conditions, which can bring about energy shortages, climate warming, and environmental damage [4]. Therefore, the development of carbon materials that are environmentally friendly, low cost, widely available, and renewable has been a continuous research direction of exploration.

Biochar, a carbon by-product obtained from natural resources or biomass (e.g., plants [5], animals [6], insects [7,8], etc.) after carbonization, is a new cost-effective and environmentally preferable degradable biomaterial. Biochar also has great strengths over conventional carbon materials such as graphite and carbon nanotubes in terms of production capacity, resource reserves, cost expenditures, and environmental protection [9,10]. In addition, biochar can greatly reduce the use of fossil fuels and help with sustainable development [11]. Thermal processes such as hydrothermal carbonization [12,13], pyrolysis [14,15], microwave irradiation [16,17], and laser processing [18,19] have been used to convert biomass precursors into biochar. In addition, due to its porosity, functional groups, electrical conductivity, and biocompatibility, biochar has been widely investigated and applied in the fields of wastewater treatment [20], capacitors [21], sensors [22], electrochemistry [23], and environment [24], making it a hot research topic today (Figure 1a). Recently, the application of biochar in the printing field has also attracted the attention of researchers (Figure 1b).

Inks used in screen printing and flexographic printing have the potential to be replaced by biochar, according to research from previous years [25,26]. Additionally, the development of high-performance 3D printing polymer blends and composites is urgently needed due to the limits of pure polymers in terms of printability and mechanical qualities. Researchers have found that the use of artificially generated plastics can be combined with sustainable fillers, such as biochar, to meet the demands of intended applications [27,28,29,30]. Biochar also has many potential applications in treating printing and dyeing wastewater [31,32,33,34]. In this paper, we first describe the typical carbonization methods and modifications used in the preparation of biochar, and then summarize their research in the printing field.

## 2. Raw Material and Preparation Method of Biochar

The carbonation of biomass is an elemental change process in which most of the O, H, and other elements of biomass resources disappear and the content of C elements is significantly increased. Biomass resources are made up of intricate parts, which have various pyrolysis methods. As a result, choosing the right biomass precursors is crucial to producing biochar with the desired structure and morphology [35]. In general, the sources of biochar are classified as plant-derived biomass and animal-derived biomass (Figure 2). In past studies, plant-derived biomass has been widely used as precursors of biomass carbon, e.g., coffee grounds [36,37], rice husk [38], tea grounds [39], seaweed [40], and radish [41]. Compared to plants, animal-derived biomass contains more complex components, and chitin [42], animal hair [7], animal feces [43], and eggs [44] have been used to synthesize biochar. Dehydration, decarboxylation, depolymerization, isomerization, aromatization, and carbonization are some of the reactions that occur simultaneously or in sequence during the carbonization of biomass, and they all contribute to the creation of biochar with various structures and properties [45,46]. The carbonization process is incredibly crucial in the creation of biochar, in addition to the choice of precursors. The selection of the carbonization method directly affects the chemical and physical properties of biochar such as structure, surface area, porosity, chemical composition, functional groups, and graphitization [47]. Therefore, we have reviewed several typical carbonization methods to provide a reference for later studies (Table 1).

### 2.1. Pyrolysis Carbonization

In general, pyrolysis carbonation is the broad term for the thermal decomposition of raw materials using an apparatus with limited or no oxygen [48]. Tube furnaces and muffle furnaces are the more common instruments used for pyrolysis. In this process, biomass is pyrolyzed to form solids, liquids, and gases (Figure 3a). The solid and liquid products are commonly termed biochar and bio-oil, and the gaseous products often contain small-molecule chemicals including carbon dioxide, hydrogen, and carbon monoxide [49]. Pyrolysis carbonization is one of the most popular and commonly utilized techniques for producing biochar. This is because the structure and qualities of biochar can be easily regulated by controlling the carbonization temperature, heating rate, carbonization time, and gas atmosphere, and the apparatus used for pyrolysis carbonization can precisely control the above parameters [45]. However, compared to other methods, the high energy consumption of the pyrolysis process is not negligible [4].

Depending on the heating rate and carbonization time, pyrolysis carbonization can be classified into three categories: slow (conventional) pyrolysis, fast pyrolysis, and flash pyrolysis [50]. Slow pyrolysis, which was initially applied to the production of charcoal, reacts at relatively low temperatures (~300–700 °C) for long periods (lasting from hours to days). When pyrolysis is carried out at a slow heating rate, adequate time for repolymerization occurrences can increase solid yields, so slow pyrolysis results in more coke and less tar formation [51]. Fast pyrolysis is characterized by fast heating rates and brief residence durations, leading to rapid cracking of biomass, resulting in more bio-oil and less coke, with higher bio-oil yields [45]. Flash pyrolysis processes use higher heat rates (10^3^–10^4^ °C/s) and faster residence times (<0.5 s), leading to remarkably high bio-oil yields [52,53]. It can be concluded that the pyrolysis products of biomass can be regulated in terms of yield and composition by modulating the pyrolysis temperature and the pyrolysis rate. Therefore, slow heating rates are more favorable in the synthesis of biochar, and therefore, low heating rates are used in most reports to produce biochar [4].

**Figure 3 materials-16-05081-f003:**
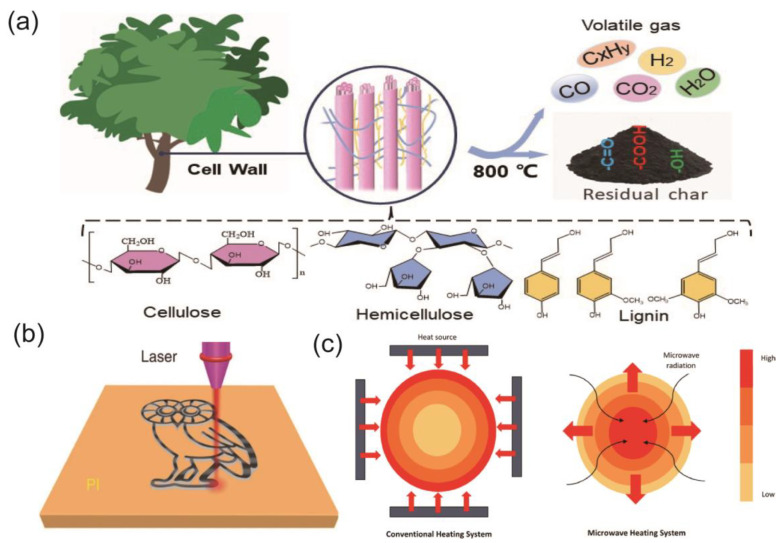
(**a**) Composition and pyrolysis products of woody biomass. (**b**) Schematic diagram of the process of PI synthesis of LIG [54] (reproduced with permission from Jian Lin, Nat. Commun.; published by Springer Nature, 2014). (**c**) Comparison of the traditional heating system and microwave heating system [55] (open access; published by Elsevier, 2021).

### 2.2. Hydrothermal Carbonization

Hydrothermal carbonization (HTC) technology is the generation of hydrated carbon from biomass or its components, using water as a solvent and reaction medium at a certain temperature [56]. During the HTC process, three types of reactions result in biomass dehydration: fragment polymerization, intermolecular dehydration, and carbonization [4]. Modulation of the structure, components, and morphology of hydrated carbon can be readily achieved by changing the conditions of the hydrothermal reaction, such as biomass precursors, temperature, and reaction time. Sun et al. [57] used glucose as a raw material to obtain carbon nanospheres with a diameter of 200 nm in a hydrothermal reaction at 160 °C for 3.5 h. Chen et al. [58] used low-temperature (L-T) HTC technology to reform waste sludge into biochar. Biochar with a maximum high heating value of 15.21 MJ/kg was obtained, and the product was similar to cellulose biochar and lignite surface. In addition, some xanthic and humic acid analogs were observed in the secondary leachate of carbon, which could promote redox reactions in the soil microenvironment. This illustrates that L-T HTC is a good method for sludge treatment and resource recovery.

HTC has many advantages: (1) compared to pyrolytic carbonation, the feedstock for HTC is allowed to be in an incomplete dry state and does not need to undergo a dehydration process before the reaction [59]; (2) compared to other carbonation methods, HTC requires lower temperatures, reducing the energy loss in the carbonation process [60]; (3) because of the relatively mild reaction conditions of HTC, the yield of solid products is commonly higher; (4) the obtained liquid and gas phases could be value-added after further processing [61]. In addition, hydrochar, with its high adsorption capacity and no negative effects on plants, has attracted much attention for its application in soil amendment, increasing carbon stocks in degraded soils, and pollutant remediation [62]. Hydrothermal carbonization has also shown great potential in converting biomass waste into solid biofuels [60]. Unfortunately, the lower pyrolysis temperature also brings the problem of poor graphitization, which limits the application of hydrochar [4].

### 2.3. Laser-Induced Carbonization

In 2014, Tour et al. [54] discovered that commercial polyimide (PI) films could be converted into 3D porous laser-induced graphene (LIG) using a mid-infrared (MIR) CO_2_ laser system under ambient conditions (Figure 3b). Moreover, LIG demonstrates exceptional qualities like high porosity (340 m^2^·g^−1^), strong heat stability (900 °C), and superior electrical conductivity (25 S·cm^−1^). Since then, much research around laser-induced carbonization (LIC) technology has been carried out. With LIC technology, pre-designed patterns with regulated microstructure, surface characteristics, electrical conductivity, chemical composition, and heteroatom doping can be effectively manufactured on various carbon materials. In addition, the LIC technique has the benefits of being inexpensive, without chemicals, metal-free, and mask-free, reducing the consumption of initial supplies and the environmental effect [63].

So far, the raw material for LIC technology has been extended from PI to many commercial polymers and natural materials [64]. Truong et al. [65] formed highly conductive graphene and high-resolution LIG patterns on a variety of woods and leaves in ambient air via programmable irradiation with high-repetition-rate UV FS laser pulses in one step. New possibilities are offered for flexible green electronics. In addition, the laser used in this work has a lower input power (0.8 W) than the CO_2_ laser (7.8 W) but is compared to the reported conductivity of LIG on wood with higher laser power under an inert atmosphere or multi-step strategy. In addition, Morosawa et al. [66] fabricated graphitic carbon on cellulose nanofiber (CNF) films using a high-repetition femtosecond laser. In the case of laser irradiation, the thermal effect carried by heat buildup and the optical influence brought on by the high peak intensity created at the focal point contributes to the degradation of CNF, leading to the production of crystalline graphitic carbon. One scan of the laser beam yielded a conductivity of up to 6.9 S/cm, which is more than 100 times greater than previously reported. LIG not only has the excellent properties of graphene, but also has the advantage of controlled performance and easy production, and has been found to have good applications in numerous fields such as biosensors, supercapacitors, batteries, and frictional electric nanogenerators prospect [63]. On the downside, this method can only achieve surface and localized carbonization and is mostly performed in the laboratory [4].

### 2.4. Microwave-Assisted Carbonization

A rapidly expanding area of materials research, microwave-assisted synthesis, and processing has effectively entered the realm of carbon nanomaterials [67]. The mechanism of microwave-assisted carbonization (MAC) is to transform electromagnetic radiation into heat within the irradiated things. During the preparation process, there is no actual contact between the biomass and the heat source; instead, microwaves enter the sample and the energy is transformed to heat inside the particles [68]. The transfer of the energy causes a temperature gradient within the particle from the interior to the exterior, which is the same as the direction of the volatiles emitted [69]. Pyrolysis of biomass in MAC is easier and faster compared to other carbonization methods, thus showing clear advantages in providing a fast, energy-efficient, and pre-designed heating process that helps to increase productivity and reduce production costs [70]. Haeldermans et al. [71] thermally treated medium-density fiberboard using two different carbonization methods (MAC and pyrolytic carbonization) (Figure 3c). The results showed that MAC produced biochar with higher aromaticity at lower temperatures than pyrolytic carbonization for the same input material.

However, MAC also has some drawbacks. For example, due to the lack of standards for instrument operation and testing techniques, the experimental results are not reproducible. In addition, “thermal runaway” and “hot spots” may occur during MAC, and the lack of a theoretical basis for MAC–material interactions makes the large-scale application of this system a great challenge [55]. In general, there is a wide range of methods for preparing and modifying biochar, each with its advantages and disadvantages, and a suitable method needs to be chosen according to the requirements of the application.

## 3. Modification of Biochar

The properties of the original biochar are not sufficient to satisfy the needs of scientific research and practical production, which limits the development of biochar applications. Therefore, some additional methods can be adopted to improve the properties of biochar such as pore size distribution, thermal properties, and specific surface area. In general, most of the modification methods of biochar revolve around physical and chemical methods.

### 3.1. Physical Modification

Physical modification generally does not involve chemical reagents, which is simple and economical, and is a common means of biochar modification (Table 2). The advantage of physical modification consists in improving the pore structure and oxygen-containing functional group induction of biochar in a cost-effective and safe way [72]. Due to the increase in specific surface area and pore volume, the adsorption capacity of biochar may be improved for heavy metal elements, nutrients, and organic pollutants [73]. Increasing reaction rates and reducing energy consumption are problems that should be addressed [74]. Ball milling, steam activation, etc., are typical physical modification methods.

Ball milling is a conventional treatment method to grind biomass into smaller particles by physical means. In this way, the heat transfer during pyrolysis is facilitated, creating a uniform temperature within the particles and thus increasing the yield of bio-oil by inhibiting char formation and secondary cracking by steam [45]. Steam activation is usually carried out at 500–850 °C, with treatment times ranging from 1 to 7 h, and is an effective method for modifying biochar properties (specific surface area and porosity) [75]. Steam activation is usually performed during the initial pyrolysis, where the prepared biochar is exposed to steam for partial vaporization. This step promotes the formation of crystalline C and the partial degreasing of the biochar [76]. With the assistance of steam, biochar could develop new pores and expand small pores formed during pyrolysis [77].

**Table 2 materials-16-05081-t002:** Various raw materials and physical modification methods.

Raw Materials	Modification Methods	Heating Temperature (°C)	Surface Area (m^2^·g^−1^)	Pore Volume (cm^3^·g^−1^)	Application	Reference
Burcucumber	Steam	300 and 700	7.10	0.038	Remove sulfamethazine	[78]
Tea waste	Steam	300 and 700	576.09	0.1091	Remove sulfamethazine	[79]
Corncobs	Thermal air oxidation	300–700	~350	-	-	[80]
Softwood	Steam	400	672	-	Adsorb tetracycline	[81]
Wheat straw	Ball milling	400–800	271.10	0.1445	Adsorb volatile organic compounds	[82]
*Danshen*	Grind	250–800	70.3	0.068	Adsorb sulfamethoxazole	[83]

### 3.2. Chemical Modification

Chemical modifications usually involve the addition of chemical substances to the feedstock to change the surface chemistry of the biochar (Table 3). Many chemical reagents have been used for biochar modification, such as acids (HCl, H_2_SO_4_, H_3_PO_4_, and HNO_3_), bases (KOH, K_2_CO_3_, and NaOH), salts (chlorides and phosphates), and oxidants (KMNO_4_ and H_2_O_2_) [84]. Compared to physical activation, chemical activation has the advantage of fast reaction time and low operating temperature, but chemical modification costs more, and the strong bases, acids, and salts used to lead to secondary contamination. However, in recent years, researchers have discovered organic acids that are more friendly to the environment and have been widely used as surface chemicals for modified biochar [85]. Sun et al. [86] extracted biochar from eucalyptus sawdust and then modified eucalyptus charcoal using citric acid, tartaric acid, and acetic acid at low temperatures and used them as adsorbents to remove methylene blue (MB) from contaminated water. Experimental data showed that citric-acid-modified eucalyptus char had higher MB adsorption efficiency than tartaric- and acetic-acid-modified eucalyptus char. In addition, more gentle modifiers such as hydrogels, vitamins, and chitosan have been used in biochar in recent years [87].

## 4. Application of Biochar in the Printing Field

The unique chemical property structure, low cost, huge reserves, and environmental friendliness make biochar a potential for a wide range of applications in wastewater treatment [98], capacitors [99], sensors [100], and the environment [101]. Printing technology, which has a very long history in China, has also achieved extraordinary technological innovations in recent years that have changed and facilitated our lives. However, with the growing awareness of sustainable development, researchers are constantly trying to move the raw materials, manufacturing processes, and by-products of printing in a greener direction. In recent years, some researchers have applied biochar to the printing field, and it has shown good application potential. But a systematic summary is lacking; therefore, this paper reviews the application of biochar in the printing field.

### 4.1. Biochar as Printing Material

In recent years, there has been a growing interest in the sustainability of printing inks to reduce the carbon footprint of the printing industry. Screen printing (SPE) is a well-established technology used in the mass production of electrochemical sensors [102]. Due to its inert internal structure and highly functionalized surface, biochar offers excellent electron transfer kinetics, reproducibility, and high sensitivity [103]. In addition, the structural characteristics of biochar show certain similarities to carbon nanomaterials (i.e., graphene, nanotubes, and nanofibers), which are widely used in electrochemistry, and therefore have the potential to be a renewable and biodegradable raw material for screen-printing [104]. In addition, inks used in printing technology mainly consist of colorants, additives, carriers, and solvents [105]. Carbon black (CB), the main raw material for black pigments in printing inks, is obtained from the incomplete combustion of natural gas, thus becoming one of the “culprits” of greenhouse gas emissions. With the efforts of researchers, “green” inks using biochar instead of conventional CB have also been developed. This paper summarizes some of the more typical efforts.

Cancelliere et al. [106] prepared biochar-based SPE electrodes using biochar prepared from waste grains of breweries as raw material. To investigate its electrochemical potential and performance, different electroactive substances were used, and the results were compared with the electrodes produced with commercial graphene, which were found to exhibit better electrochemical behavior in resolution, peak-to-peak separation, current intensity, and resistance to charge transfer (Figure 4a). In the follow-up study, they also proposed a reverse-designed biochar-based SPE electrode with higher sensitivity and reproducibility compared with conventional SPE [103]. Goh et al. [25] formulated flexographic inks to replace CB with a more sustainable pine char. The results showed that prints with a reflective optical density of more than 1.0 were obtained with excellent tonal reproduction (Figure 4b). Yuan et al. [107] also developed a stabilized biochar-based ink made from fruit peels that can be integrated on different flexible substrates by SPE or program-controlled writing methods. This suggests that replacing fossil fuel charcoal with biochar for printing materials is a promising avenue, but more research is still needed to lay the theoretical foundation.

### 4.2. Application of Biochar in 3D Printing

In recent years, polymeric nanocomposites with carbon-based reinforcing fillers such as carbon nanotubes, graphene, and carbon black have been widely used to improve the properties of additive manufacturing (3D printed) composites [108,109]. The precursors of these materials are dependent on fossil fuels and may be in crisis in the future, which further aggravates environmental problems such as the energy crisis, global warming, and environmental pollution. In this context, there is a great demand for the development of carbon with mass production potential and simple synthesis processes as sustainable precursor materials for 3D printing. Commercially mature and low-cost biochar shows great potential for applications [110,111]. Biochar has the advantages of high hardness, high surface area, and good thermal properties, and the thermal and mechanical properties of materials such as polylactic acid (PLA), epoxy resin, thermoplastic polyurethane (TPU), and polyethylene terephthalate (PET) can be improved by adding biochar as a filler in polymers [112,113,114].

Umerah et al. [115] prepared a carbon-based biodegradable polymer nanocomposite using coconut shell carbon and PLA. The results showed that the addition of coconut shell carbon improved the mechanical and electrical properties of the material and was successfully used in 3D printing (Figure 5a). Mohanad et al. [29] prepared a composite material using biochar and PET, which could be used in 3D printing. The results demonstrated that adding biochar enhanced the mechanical, thermal, and kinetic properties of the composites. The tensile strength of PET increased by 32% after receiving a 0.5-weight percent infusion of biochar. The tensile modulus of polymer composites loaded with 5 wt.% increased by 60% compared to pure PET. Mohammed et al. [116] synthesized biochar by heating biodegradable starch-based packaging trash (Figure 5b). In addition, to make the produced biochar more suited for dispersion and 3D printing, the obtained biochar was sonicated using ultrasounds to reduce the particle size and enhance the active surface area. The biochar was used as an adequate strengthening filler for polypropylene polymers, increasing the strength and tensile modulus by 46% and 34%, respectively, with only 0.75 wt.% loading. All these works show that biochar has good prospects for 3D printing applications.

### 4.3. Biochar as Adsorbent

Printing and dyeing industries are one of the factors that cause ecological damage because they discharge large amounts of dyes in their wastewater. Printing and dyeing wastewater is not only huge in quantity, but also has the problems of large amounts of residual dyes, poor biodegradability, and high alkalinity, making it difficult to treat [117]. Therefore, efficient dye removal is an urgent problem in dye wastewater remediation nowadays [31]. Among the many physical and chemical methods for dye removal, adsorption has become an effective method because of its advantages of high selectivity, low cost, and simple operation [118]. Biochar shows great potential for dye adsorption due to its large specific surface area, porous structure, abundance of surface functional groups, and low cost [119,120] (Figure 6). However, pristine biochar usually lacks excellent adsorption properties, and the use of physical or chemical modifications to improve the surface groups and porous structure of biochar can significantly increase its adsorption capacity for removing pollutants from wastewater [121,122].

Chen et al. [123] used corn stover carbon activated by K_2_CO_3_ for the decolorization of printing wastewater and investigated the adsorption of methylene blue (MB) and gentian violet (GV). In the one-component system, the equilibrium adsorption of MB was 274.84 mg·g^−1^, which was higher than that of GV at 266.57 mg·g^−1^, but in the binary system, the GV improved the adsorption of MB up to 325.15 mg·g^−1^ and 287.73 mg·g^−1^ at different GV concentrations. Huo et al. [32] prepared N-self-doped mesoporous lotus leaf biochar (LLC800) from lotus leaves to degrade dyeing and printing wastewater. The LLC800/peroxynitrite (PS) system achieved 99.46% removal of AO7 from dyeing and printing wastewater. These studies have tapped into the application potential of biochar and provided more value-added application scenarios.

## 5. Limitations and Future Perspectives

Biochar is a unique renewable resource with the advantages of low cost, huge reserves, and environmental friendliness, and it can also be extremely easily endowed with new properties using physical or chemical modification. Although biochar has received much attention in the printing field in recent years, its application is mainly in the laboratory, and there are still many challenges in transferring it from the laboratory scale to large-scale application. For example, although raw materials for biochar are widely available, the quality and properties of biochar vary depending on the type of feedstock and pyrolysis conditions, posing some obstacles to large-scale production and applications. For example, the quality of biochar is more difficult to control than other 3D printing materials, which makes it more difficult to produce the same product accurately and reproducibly. When used as a replacement for CB, it also poses problems for mass production because of the inconsistent quality. Therefore, the preparation of biochar must regulate its performance more precisely to further ensure that biochar possesses a more stable quality.

In addition, the pyrolysis process, which devastates the structure of the raw material of biochar, leads to poor mechanical properties of biochar, usually several times lower than traditional plastic and resin materials, making it unsuitable for high-strength applications and mostly present as an additive in 3D printing applications. Although there are some drawbacks in the application and preparation of biochar, its good economic advantages, great application potential, huge reserves, and sustainability are not negligible. If it can be reasonably applied to replace some petroleum materials, it would be a great effective strategy to solve energy and environmental problems. However, research on the application of biochar in printing is still very limited, and more extensive and detailed studies are needed to lay the theoretical foundation.

## 6. Conclusions

In general, with the increasing awareness of environmental protection and the urgent need for global emission reduction, the application of biochar in the printing field will become more and more promising. According to the current research trend, biochar will be more widely used, including but not limited to ink manufacturing, printing, dyeing wastewater treatment, manufacturing 3D printing materials, etc. With the continuous development of science and technology, the production technology and quality of biochar will be improved, and the application scope and practicality will be more extensive. Although, up to now, the application of biochar in the printing field is mostly in the laboratory, there are still some limitations and technical defects in the large-scale application. We hope that through the efforts of scientific researchers, more relevance will be given to the application of biochar in the printing field soon. This paper summarizes the preparation, modification, and application of biochar in the printing field, to provide readers with more references and knowledge of this field.

## Figures and Tables

**Figure 1 materials-16-05081-f001:**
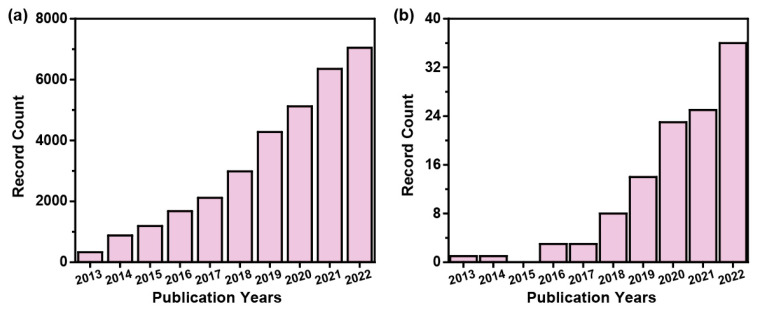
(**a**) The number of publications on biochar (data from Web of Science for the period from 2013 to 2023, using “biochar” as a topic), and (**b**) the number of publications (data from Web of Science for the period from 2013 to 2023, using “biochar” and “print” as topics).

**Figure 2 materials-16-05081-f002:**
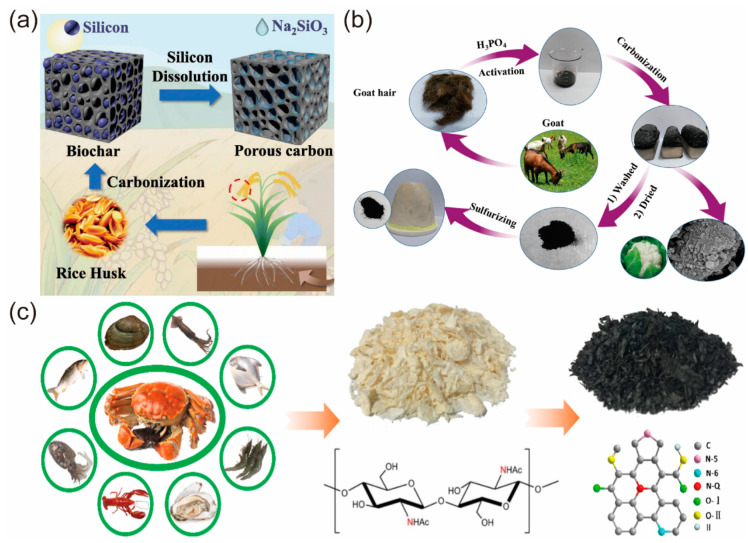
(**a**) Preparation process of rice husk carbon [38] (reproduced with permission from Fengli Gan, Total Environ.; published by Elsevier, 2021). (**b**) P-doped goat hair porous carbon [7] (reproduced with permission from Juan Ren, J. Energy Chem; published by Elsevier, 2019). (**c**) Schematic diagram of N-doped carbon from bio-waste chitin [42] (reproduced with permission from Rui Hao, *Nano Energy*; published by Elsevier, 2018).

**Figure 4 materials-16-05081-f004:**
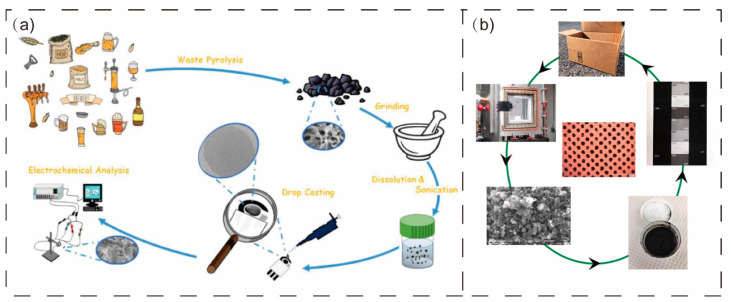
Brewery waste grain carbon (**a**) (open access; published by MDPI, 2019) and pine wood carbon (**b**) in printing [25,106] (reproduced with permission from Yang Goh, J. Cleaner Prod; published by Elsevier, 2021).

**Figure 5 materials-16-05081-f005:**
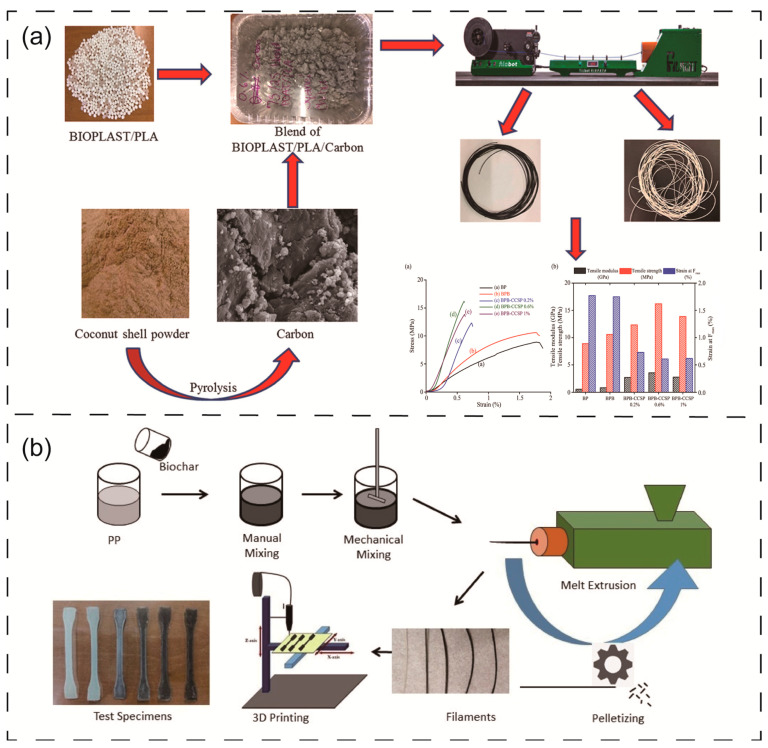
(**a**) Preparation process of using coconut shell carbon in 3D printing [115] (reproduced with permission from Chibu O. Umerah, Composites, Part B; published by Elsevier, 2020). (**b**) Starchy carbon [116] in 3D printing (open access; published by Elsevier, 2022).

**Figure 6 materials-16-05081-f006:**
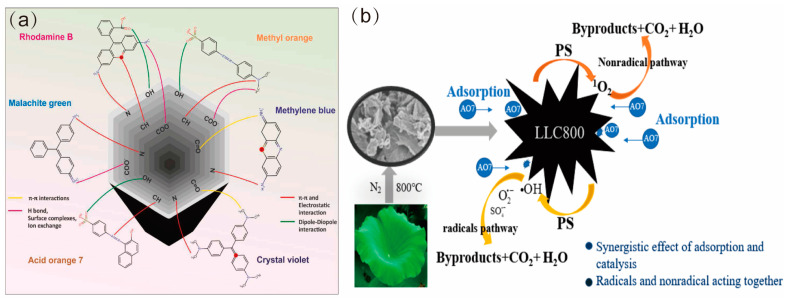
(**a**) Combination of biochar with different dyes [31] (reproduced with permission from Shubham Sutar, Environ Res; published by Elsevier, 2022). (**b**) Degradation of AO7 by LLC800/PS system [32] (reproduced with permission from Jiaxu Huo, *Catalysts*; published by MDPI, 2020).

**Table 1 materials-16-05081-t001:** Advantages and disadvantages of different preparation methods.

Preparation Methods	Advantages	Disadvantages
Pyrolysis carbonization	Controllable carbonization temperature Controllable heating rate Controllable gas atmosphere	High energy consumption Long reaction process
Hydrothermal carbonization	Without pre-drying process Low reaction temperatures Mild reaction conditions Low energy consumption	Low graphitization degree Long reaction process
Laser-induced carbonization	High porosity Superior electrical conductivity Controllable performance Fast reaction process	Surface carbonization Localized carbonization Mostly performed in the laboratory
Microwave-assisted carbonization	Fast reaction process Energy-efficient	Difficult to control reaction temperature Poor reproducibility

**Table 3 materials-16-05081-t003:** Various raw materials and chemical modification methods.

Raw Materials	Modification Methods	Heating Temperature (°C)	Surface Area (m^2^·g^−1^)	Pore Volume (cm^3^·g^−1^)	Application	Reference
Rice straw	HCl	700–1000	2356	1.61	-	[88]
Rice Husk	KOH	700	403.0	0.35	CO_2_ Capture	[89]
Poplar chips	AlCl_3_	550	418.14	0.413	Adsorb NO^3−^ and PO4^3−^	[90]
Macroalgae	H_2_SO_4_ and NaOH	450	45.463	0.0318	Adsorb phosphate	[91]
Waste poplar leaves	Urea	650	-	-	Adsorb Cr(VI)	[92]
Mangosteen shells	HCl, KOH and ZnCl_2_	350, 700	1836.46	1.058	Adsorb Cr(VI)	[93]
Swine manure	H_3_PO_4_	700	372.21	0.23	Adsorb tetracycline	[94]
Pinecone	Zn(NO_3_)_2_·6H_2_O	500	11.54	0.028	Remove Arsenic(III)	[95]
Biochar	H_2_SO_4_/KOH	60 °C stirrer	117.8	0.073	Remove tetracycline	[96]
Rice husk	KOH	500	3263	-	Supercapacitor	[97]

## Data Availability

The authors confirm that the findings of this study are available within the article.

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
