# Peer review of "Preparation, Modification, and Application of Biochar in the Printing Field: A Review"

_materials, 2023, doi:10.3390/ma16145081_

Round 1
Reviewer 1 Report
Manuscript Number: materials-2457646
Title: Preparation, modification, and application of biochar in the printing field: a review
Article Type: review article
In the manuscript a review dedicated to the biochar is presented. Authors focused on the production of the biochar, its modification and properties enhancement and its application in the printing field. The manuscript is well written. Everything is clearly presented. The topic is interesting and fits well into the Journal’s scope. In their work Authors refer to 135 sources from which almost half is from the last three years.
The biggest advantage of the manuscript is the review of the most current publications dedicated to the topic of biochar. The biggest disadvantage of the manuscript is a small number of references dedicated to the printing applications – only 14 references (although Fig. 1 shows that Authors found about 100 such publications).
Below I am presenting my remarks:
1. Figure 2 caption should be corrected (capital letter).
2. Line 164: It is “reduced input force” it should be “reduced input power”
3. Figure 5 is too small.
4. Line 291: Please expand the abbreviation PLA.
5. Line 293: Please correct the following sentence: “Mohanad et al. [54] were prepared from biochar with polyethylene terephthalate (PET) into composites that can be used for 3D printing.”
6. Line 301: It is “obtained biochar was ultrasound to reduce” it should be “obtained biochar was sonicated using ultrasounds to reduce”
7. Line 310: “Application of biochar in printing and dyeing wastewater” I'm not quite sure what this phrase means. Can authors express it differently?
8. References which have to be corrected:
[1] please give more information about this source. Is it a book? A chapter in book? A paper? Conference paper?
[4] DOI is missing
[8] DOI should be corrected
[10] DOI should be corrected
[13] DOI should be corrected
[14] DOI should be corrected
[18] DOI should be corrected
[28] DOI should be corrected
[29] DOI is missing
[30] DOI should be corrected
[33] DOI is missing
[36] DOI should be corrected
[37] DOI should be corrected
[39] DOI should be corrected
[44] DOI should be corrected
[46]DOI should be corrected
[50] Please give location and country
[52] DOI should be corrected
[53] DOI should be corrected
[54] DOI should be corrected
[55] DOI should be corrected
[57] DOI should be corrected\[58] DOI should be corrected
[60] DOI should be corrected
[62] DOI should be corrected
[71] Pages and DOI are missing
[73] vol, pages and DOI are missing
[74] DOI should be corrected
[75] DOI is missing
[76] DOI should be corrected
[77] DOI should be corrected
[78] DOI should be corrected
[80] DOI should be corrected
[84] DOI should be corrected
[86] DOI should be corrected
[88] DOI should be corrected
[90] DOI should be corrected
[91] DOI should be corrected
[92] DOI should be corrected
[93] DOI should be corrected
[94] DOI should be corrected
[97] DOI should be corrected
[100] What kind of publication is it? Book? Chapter? Paper? Key information are missing
[103] DOI should be corrected
[115] DOI should be corrected
[119] DOI should be corrected
[121] DOI should be corrected
[122] DOI should be corrected
[123] DOI should be corrected
[124] DOI should be corrected
[126] DOI should be corrected
[127] DOI should be corrected
[129] DOI should be corrected
[132] DOI should be corrected
[133] DOI should be corrected
[134] DOI should be corrected
Some sentences should be corrected.
Reviewer 2 Report
At first glance, the article seems to be a good summary. It summarises the role of biochar and its production potential in a clear and purposeful way. It would have been very useful to compare the energy requirements of the different production methods, although this is referred to in several places elsewhere in the article.
However, the biochar modification chapter raises more and more questions and gaps. For example, the table 1. summarising the possible modifications should be supplemented with a column on what was the aim of the modification or what was achieved by it? (pore size/porosity/specific surface area change?)
Compared to the last 5 years cc. 120 articles have been published on the use of biochar in printing (see Figure 1), the chapter on biochar in printing contains only cc. 20 articles, although the title suggests that this is the main topic of their work. A more in-depth review of the literature is needed, e.g. what properties biochar must have to be suitable for use in the printing industry/PC printers/3D.
Reviewer 3 Report
dear author,
it is a good paper and gives a good overview of biochar especially the production; the application of printing is a bit short.
I only have one recommendation:
Table 1 includes both physical and chemical modification
Please divide the table into two tables (physical and chemical). Place the second table after the chemical modification section.
BR
The reviewer
Reviewer 4 Report
materials-2457646
The manuscript by Li et al. “Preparation, modification, and application of biochar in the printing field: a review” briefly summarizes the typical preparation and modification methods of biochar for their potential application in the printing field with perspectives. Overall, the manuscript requires major revision before its publication in “materials” as follows:
Comments:
1. Lines 27-28, also add carbon materials as carbon particles and carbon scaffolds i.e., ACS Appl. Mater. Interfaces 2019, 11, 18968−18977.
2. The details of various preparation methods of biochar should be presented as a Table with their advantages and disadvantages (each method).
3. Section 4 can be majorly revised with detailed quantitative information. Also, it can be extended with other minor sub-sections on “other potential applications of biochar in the related area”.
4. Table 1 can be extended with the details of the application.
Moderate changes are required.
Round 2
Reviewer 2 Report
The corrections and changes have complemented the article well, so it is now of professional interest and reference.
Reviewer 4 Report
Accept